# Multi-Scale Representation Learning for Protein Fitness Prediction

**Zuobai Zhang**[1,2,*]  **Pascal Notin**[3,*]  **Yining Huang**[3]  **Aurélie Lozano**[5]
**Vijil Chenthamarakshan**[5]  **Debora Marks**[3,4]  **Payel Das**[5,†]  **Jian Tang**[1,6,7,†]
[*]equal contribution      [†]corresponding author
[1]Mila - Québec AI Institute,    [2]Université de Montréal,    [3]Harvard Medical School,
[4]Broad Institute,    [5]IBM Research,    [6]HEC Montréal,    [7]CIFAR AI Chair
`zuobai.zhang@mila.quebec`,  `pascal_notin@hms.harvard.edu`,
`daspa@us.ibm.com`,  `jian.tang@hec.ca`

## Abstract

Designing novel functional proteins crucially depends on accurately modeling their fitness landscape. Given the limited availability of functional annotations from wet-lab experiments, previous methods have primarily relied on self-supervised models trained on vast, unlabeled protein sequence or structure datasets. While initial protein representation learning studies solely focused on either sequence or structural features, recent hybrid architectures have sought to merge these modalities to harness their respective strengths. However, these sequence-structure models have so far achieved only incremental improvements when compared to the leading sequence-only approaches, highlighting unresolved challenges effectively leveraging these modalities together. Moreover, the function of certain proteins is highly dependent on the granular aspects of their surface topology, which have been overlooked by prior models. To address these limitations, we introduce the Sequence-Structure-Surface Fitness (**S3F**) model — a novel multimodal representation learning framework that integrates protein features across several scales. Our approach combines sequence representations from a protein language model with Geometric Vector Perceptron networks encoding protein backbone and detailed surface topology. The proposed method achieves state-of-the-art fitness prediction on the ProteinGym benchmark encompassing 217 substitution deep mutational scanning assays, and provides insights into the determinants of protein function. Our code is at `https://github.com/DeepGraphLearning/S3F`.

## 1 Introduction

Proteins carry out a diverse range of functions in nature – from catalyzing chemical reactions to supporting cellular structures, transporting molecules or transmitting signals. These functions are uniquely determined by their amino acid sequences and three-dimensional structures. The ability to design these sequences and structures presents significant opportunities to tackle critical challenges in sustainability, new material, and healthcare. This optimization process typically begins by learning the relationship between protein sequences or structures and their function, referred to as a *fitness landscape*. This multivariate function describes how mutations impact protein fitness – the more accurately we model these landscapes, the better we can engineer proteins with desired traits [1, 2].

A significant challenge in modeling the fitness landscape is the scarcity of experimentally collected functional labels relative to the vastness of protein space [3]. As a result, self-supervised approaches to protein representation learning have become crucial for predicting mutation effect [4, 5]. While initial methods focused on learning a family-specific distribution over homologous protein sequences

retrieved with a Multiple Sequence Alignment [4, 6–9], subsequent methods have sought to learn general functional patterns across protein families, giving rise to 'protein language models' or 'family-agnostic models' [10–12]. Recently, hybrid models have achieved state-of-the-art fitness prediction performance by leveraging the relative strengths of both types of approaches [13–15].

Although sequence-based methods are effective in recapitulating certain aspects of protein structure [16, 13], several protein functions and tasks crucially benefit from using a more granular representation of the protein structures and surfaces [17, 18]. To bridge this gap, recent studies have exploited advances in protein structure representation learning [19–21]. For instance, inverse folding models that learn a distribution over protein sequences conditioned on a protein backbone have shown enhanced performance in stability prediction [21–25]. Recent efforts have also focused on integrating sequence-based and structure-based approaches [26, 27]. A prominent example, AlphaMissense, employs structural prediction losses to distill structural information into a hybrid model, highlighting the value from structural features [28]. However, these hybrid sequence-structure methods have thus far only achieved modest improvements over leading sequence-based models, or have not made their model weights publicly available. Furthermore, current methodologies fall short in effectively modeling protein surfaces, which are essential for deciphering protein interactions and capturing broader structural details [29].

In this work, we introduce a multi-scale protein representation learning framework that integrates comprehensive levels of protein information for zero-shot protein fitness prediction (Fig. 1). We begin with a **S**equence-**S**tructure **F**itness Model (**S2F**) by combining a protein language model with a structure-based encoder. S2F utilizes the output of the protein language model as node features for a structure encoder, specifically a Geometric Vector Perceptron (GVP) [19], which enables message passing among spatially close neighborhoods. Building on this, we develop a **S**equence-**S**tructure-**S**urface **F**itness Model (**S3F**), which enhances S2F by adding a protein surface encoder that represents surfaces as point clouds and facilitates message passing between neighboring points. These multi-scale protein encoders are pre-trained using a residue type prediction loss on the CATH dataset [30], enabling zero-shot prediction of mutation effects.

Our methods are rigorously evaluated using the comprehensive ProteinGym benchmark [31], which includes 217 substitution deep mutational scanning assays and over 2.4 million mutated sequences across more than 200 diverse protein families. Our experimental results show that S2F achieves competitive results with prior methods, while S3F reaches state-of-the-art performance after incorporating surface features. When further augmented with alignment information, our method improves the current state-of-the-art by 8.5% in terms of Spearman's rank correlation. Additionally, our methods have substantially fewer trainable parameters compared to other baselines, reducing pre-training time to several days on commodity hardware. Being both lightweight and agnostic of the model used to obtain initial node embeddings, they can be readily adapted to augment forthcoming, more advanced protein language models. To better understand the impact of multi-scale representation learning, we perform a breakdown analysis on different types of assays. Our results demonstrate the consistent improvements from multi-scale learning and show that incorporating structure and surface features can potentially correct biases in sequence-based methods, enhance accuracy in structure-related functions, and improve the ability to capture epistatic effects. We summarize our contributions as follows:

- We develop a general and modular framework to learn *multi-scale* protein representations (§ 3);

- We introduce two instances of this framework – *S2F* (§ 3.3) and *S3F* (§ 3.4), augmenting protein language model embeddings with structure and surface features for superior fitness prediction;

- We thoroughly evaluate our methods on the 217 assays from the ProteinGym benchmark, demonstrating their state-of-the-art performance and fast pre-training efficiency (§ 4.2);

- We perform a breakdown analysis for different types of assays to deep dive into the determinants of functions enabled by our multi-scale representation (§ 4.3).

## 2   Related Work

**Protein Representation Learning.** Previous research in protein representation learning has explored diverse modalities including sequences, multiple sequence alignments (MSAs), structures, and surfaces [32, 13, 20, 33]. Sequence-based methods treat protein sequences as a fundamental

biological language, employing large-scale pre-training on billions of sequences to capture complex biological functions and evolutionary signals [34, 35, 11]. Alignment-based approaches, such as MSA Transformer [13], enhance representations by incorporating MSAs, improving the capture of evolutionary relationships. Recent advancements in structure prediction tools have shifted focus toward explicitly using protein structures for representation learning [36, 20, 19, 37]. These methods employ diverse self-supervised learning algorithms like contrastive learning, self-prediction, denoising, and masked structure token prediction to train structure encoders [20, 38, 39, 26]. Additionally, extracting features from protein surfaces has shown promise in uncovering critical chemical and geometric patterns important for biomolecular interactions [29, 40, 41]. The growing availability of diverse protein data has also spurred the development of hybrid methods that combine multiple modalities for a holistic view [42, 43, 33, 44, 45]. Despite these advances, direct application of these models to zero-shot protein fitness prediction is still challenging and requires further design.

**Protein Fitness Prediction.** Learning a fitness landscape has traditionally been approached as a discriminative supervised learning task, where models are trained to predict specific targets using labeled datasets [46–49]. Recently, unsupervised fitness predictors has shown promise in surpassing these traditional methods by overcoming the limitations and biases associated with sparse labels. These unsupervised models, often designed as protein language models, are trained on vast evolutionary datasets comprising millions of protein sequences, aiming to capture a general distribution across all proteins [50, 10, 14, 11, 51]. In contrast, alignment-based models focus on specific protein families, learning from multiple sequence alignments (MSAs) to capture nuanced distribution patterns [4, 6, 8, 7, 9]. Additionally, hybrid sequence models integrate broad protein information with detailed family-specific data from alignments, enhancing the robustness of log-likelihood estimations and achieving top-tier performance [13, 15, 52].

The incorporation of structural information into protein fitness prediction has marked a promising direction in the field, inspired by recent advances in structure representation learning. Based on the concept of protein language models, SaProt utilizes structure tokens from Foldseek [53] to generalize sequence-based methods to structural data [26]. However, these structure-based methods only achieve limited improvement over the best sequence models. The latest method, AlphaMissense, integrates structural prediction losses into a hybrid model, thereby enhancing predictive accuracy [28]. Nevertheless, its high performance relies heavily on fine-tuning with weak supervision on human missense variants and it still lacks public accessibility to its model weights. Moreover, to the best of our knowledge, no existing work has employed surface-based methods for fitness prediction. To fill this gap, this work aims to integrate multi-scale protein information for fitness prediction.

## 3 Method

### 3.1 Preliminary

**Proteins.** Proteins are macromolecules that form through the linkage of residues via peptide bonds. The three-dimensional (3D) structures of proteins are determined by the specific sequence of these residues. A protein with $n_r$ residues (amino acids) and $n_a$ atoms can be represented as a sequence-structure tuple $(\boldsymbol{S}, \boldsymbol{X})$. The sequence is denoted by $\boldsymbol{S} = [s_1, s_2, \cdots, s_{n_r}]$, where $s_i \in \{1, ..., 20\}$ represents the type of the $i$-th residue. The structure is represented by $\boldsymbol{X} = [\boldsymbol{x}_1, \boldsymbol{x}_2..., \boldsymbol{x}_{n_a}] \in \mathbb{R}^{n_a \times 3}$, with $\boldsymbol{x}_i$ specifying the Cartesian coordinates of the $i$-th atom. For simplification, we only consider the alpha carbon atoms and ignore the side-chain variations induced by mutations.

**Protein Fitness Landscape.** The ability of a protein to perform a specific function, often referred to as *protein fitness*, is encoded by its sequence via spontaneous folding into structures. The effects of sequence mutations on protein function form a fitness landscape, which can be quantitatively measured through deep mutational scanning (DMS) experiments [54]. Modeling these landscapes is challenging due to the complicated relationship between sequences, structures and functions.

**Problem Definition.** Unsupervised models that predict mutational effects are becoming fundamental tools in drug discovery, addressing the challenge of data scarcity. In this paper, we explore the task of zero-shot prediction of mutational effects using structural information. For simplicity, we focus solely on substitutions, leaving the study of insertions and deletions (indels) to future work.

Formally, for each DMS assay, we start with a wild-type protein $(\mathbf{S}^{\text{wt}}, \mathbf{X}^{\text{wt}})$ and generate a set of mutants by selecting specific mutation sites and randomly replacing the original residue type with

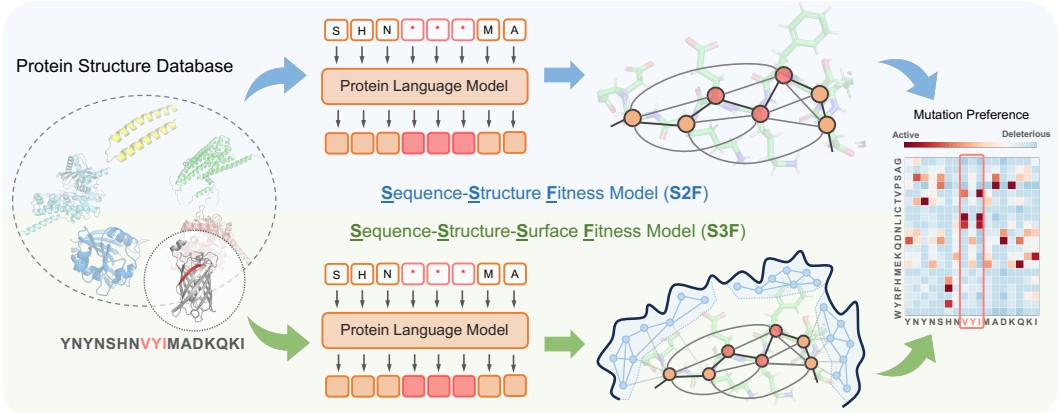

Figure 1: **Multi-scale Pre-training and Inference Frameworks for Protein Fitness Prediction**. During pre-training, protein sequences and structures are sampled from a database, with 15% of residue types randomly masked. These sequences are fed into a protein language model, ESM-2-650M. Then, the output residue representations are used to initialize node features in our structure and surface encoders. Through message passing on structure and surface graphs, our methods, **S2F** (blue) and **S3F** (green), accurately predict the residue type distribution at each masked position. This distribution is subsequently used for mutation preferences in downstream fitness prediction tasks.

a new one. For a mutant $(\mathbf{S}^{\mathrm{mt}}, \mathbf{X}^{\mathrm{mt}})$ with multiple mutations $T$, the sequence changes such that $s_t^{\mathrm{mt}} \neq s_t^{\mathrm{wt}}$ if $t \in T$; otherwise, $s_t^{\mathrm{mt}} = s_t^{\mathrm{wt}}$. We assume that the backbone structures remain unchanged post-mutation ($\mathbf{X}^{\mathrm{mt}} = \mathbf{X}^{\mathrm{wt}}$). The objective is to develop an unsupervised model that can predict a score for each mutant to quantify the changes in fitness values relative to the wild-type.

## 3.2 Protein Language Models for Mutational Effect Prediction

Protein language models trained using the masked language modeling objective are designed to predict the likelihood of a residue's occurrence at a specific position within a protein, based on the surrounding context [11, 32]. As demonstrated in [5], this method can score sequence variations using the log odds ratio between mutant and wild-type proteins for given mutations $T$:

$$\sum_{t \in T} \log p(s_t = s_t^{mt} | \boldsymbol{S}_{\backslash T}) - \log p(s_t = s_t^{wt} | \boldsymbol{S}_{\backslash T}), \tag{1}$$

where $\boldsymbol{S}_{\backslash T}$ denotes the input sequence masked at each mutated position in $T$ and an additive model is assumed over multiple mutation sites. In the zero-shot setting, inference is performed directly on the sequence under evaluation using only forward passes of the model.

## 3.3 Sequence-Structure Model for Fitness Prediction (S2F)

Although these protein language models are effective in predicting mutational effects, they do not incorporate explicit structural information during pre-training, which is crucial for determining protein functions. Building on the ESM-2-650M model, we next propose **S**equence-**S**tructure **F**itness Model (**S2F**), which integrates structural data into the sequence-based predictive framework.

A major challenge in applying structure-based methods to fitness prediction is modeling how mutations impact protein structures. To bypass this problem, we adopt a simplified assumption: the backbone structures of proteins remain unchanged post-mutation. Additionally, we choose to omit side-chain information, which reveals residue types and could potentially lead to information leakage.

**Geometric Message Passing.** We build a radius graph for each protein with nodes representing alpha carbons. Two nodes are connected if their Euclidean distance is less than 10Å. We use Geometric Vector Perceptrons (GVP) [19] to perform message passing across the graph. GVPs replace standard Multi-Layer Perceptrons (MLPs) in Graph Neural Networks, operating on scalar and geometric features that transform as vectors under spatial coordinate rotations.

We represent the hidden state of residue $i$ at the $l$-th layer by $\boldsymbol{h}_i^{(l)} \in \mathbb{R}^d \times \mathbb{R}^{d' \times 3}$ with $d$-dim scalar features and $d'$-dim vector features. The initial node feature of residue $i$ is set using the protein language model embeddings, specifically, $\boldsymbol{h}_i^{(0)} = (\text{ESM}(s_i | \boldsymbol{S}_{\backslash T}), \boldsymbol{0})$. The edge features are given by $\boldsymbol{e}_{(j,i)} = (\text{rbf}(\boldsymbol{x}_j - \boldsymbol{x}_i), \boldsymbol{x}_j - \boldsymbol{x}_i)$, where $\text{rbf}(\cdot)$ computes pairwise distance features with Radial Basis Function (RBF) kernels [55]. In the network, node and edge features are concatenated to facilitate message passing via the GVP module on the (scalar, vector) representations. Each message passing layer is followed by a feed-forward network. Formally,

$$\boldsymbol{h}_i^{(l+0.5)} = \boldsymbol{h}_i^{(l)} + \frac{1}{|\mathcal{N}(i)|} \sum_{j \in \mathcal{N}(i)} \text{GVP}\left(\boldsymbol{h}_j^{(l)}, \boldsymbol{e}_{(j,i)}\right), \tag{2}$$

$$\boldsymbol{h}_i^{(l+1)} = \boldsymbol{h}_i^{(l+0.5)} + \text{GVP}\left(\boldsymbol{h}_i^{(l+0.5)}\right), \tag{3}$$

where $\mathcal{N}(i)$ represents the set of neighbors of node $i$. The GVP module ensures SE(3)-invariance for scalar features and SE(3)-equivariance for vector features. The scalar features at the last layer $\boldsymbol{h}_i^{(L)}$ of each node $i$ are fed into a separate linear layer for predicting the residue type. Practically, we utilize $L = 5$ layers of GVP, with $d' = 16$ vector and $d = 100$ scalar hidden representations.

### 3.4 Sequence-Structure-Surface Model for Fitness Prediction (S3F)

Besides protein sequences and structures, protein surfaces—defined by neighboring amino acids—are characterized by distinct patterns of geometric and chemical properties. For instance, within a folded protein, hydrophobic residues tend to cluster inside the core, while hydrophilic residues are exposed to water solvent on its surface. These patterns provide crucial insights into protein function and potential molecular interactions. Now we integrate this aspect into our S2F model to propose a new model called **S**equence-**S**tructure-**S**urface **F**itness Model (**S3F**).

**Surface Processing.** We employ dMaSIF to generate the surface based on the backbone structure of each protein [40]. The surface is represented as a point cloud $\{\tilde{\boldsymbol{x}}_1, \tilde{\boldsymbol{x}}_2, ..., \tilde{\boldsymbol{x}}_{n_s}\} \in \mathbb{R}^3$, consisting of $n_s$ (6K-20K) points. These points are sampled based on the levels of a smooth distance function defined over each atom. Each surface point $i$ is associated with geometric features $\tilde{\boldsymbol{f}}_i$, specifically Gaussian curvatures [56] and Heat Kernel Signatures [57], providing a detailed characterization of the surface topology. In the sequel, we will use the notations $\tilde{\cdot}$ with tilde for surface points to distinguish with those for nodes in structure graphs.

**Surface Feature Initialization.** After generating protein surfaces, we build a mapping between the structure and surface graphs, projecting residue features onto the surface points. For each surface point $i$, we identify its 3 nearest neighboring residues, denoted as $\mathcal{N}_{\text{surf-res}}(i)$, each initialized with its ESM feature. These features are concatenated with their Euclidean distances to point $i$ and processed through an MLP. The average features of these neighbors are then combined with the geometric features $\boldsymbol{f}_i$. Formally, the scalar feature for surface node $i$ is initialized as follows:

$$\tilde{\boldsymbol{h}}_i^{(0)} = \text{MLP}\left(\boldsymbol{f}_i, \frac{1}{3} \sum_{j \in \mathcal{N}_{\text{surf-res}}(i)} \text{MLP}(\boldsymbol{h}_j^{(0)}, \|\tilde{\boldsymbol{x}}_i - \boldsymbol{x}_j\|_2)\right), \tag{4}$$

with the vector feature is initialized as the zero vector $\boldsymbol{0}$.

**Surface Message Passing.** To perform message passing on protein surfaces, we construct a surface graph with a k-nearest neighbor graph, wherein each surface point is linked to its 16 nearest neighbors on the surface. For each edge $(j, i)$ in this graph, the edge feature $\tilde{e}(j, i)$ is initialized using RBF kernels and represented as $\tilde{\boldsymbol{e}}(j, i) = (\text{rbf}(\tilde{\boldsymbol{x}}_j - \tilde{\boldsymbol{x}}_i), \tilde{\boldsymbol{x}}_j - \tilde{\boldsymbol{x}}_i)$.

Similar to S2F, we employ GVP to perform message passing on the scalar and vector representations:

$$\tilde{\boldsymbol{h}}_i^{(l+0.5)} = \tilde{\boldsymbol{h}}_i^{(l)} + \frac{1}{|\mathcal{N}_{\text{surf}}(i)|} \sum_{j \in \mathcal{N}_{\text{surf}}(i)} \text{GVP}\left(\tilde{\boldsymbol{h}}_j^{(l)}, \tilde{\boldsymbol{e}}_{(j,i)}\right), \tag{5}$$

$$\tilde{\boldsymbol{h}}_i^{(l+1)} = \tilde{\boldsymbol{h}}_i^{(l+0.5)} + \text{GVP}\left(\tilde{\boldsymbol{h}}_i^{(l+0.5)}\right), \tag{6}$$

where $\mathcal{N}_{\text{surf}}(i)$ represents the neighbors of surface node $i$. Similar with the approach on structure graphs, we utilize another five layers of GVP, with 16 vector and 100 scalar hidden representations.

**Residue Representation Aggregation.** After performing message passing on both the structure and surface graphs, we combine the structure representations, $\boldsymbol{h}^{(L)}$, with the surface representations, $\tilde{\boldsymbol{h}}^{(L)}$.

For each residue $i$, we identify the 20 nearest surface points, denoted as $\mathcal{N}_{\text{res-surf}}(i)$, and compute their mean representations to enhance the residue's representation. Here we use more neighbors for mapping the surface points back to residues, as there are typically much more surface points than residues. Specifically, we update each residue representation as follows:

$$h_i^{(L)} \leftarrow h_i^{(L)} + \frac{1}{20} \sum_{j \in \mathcal{N}_{\text{res-surf}}(i)} \tilde{h}_j^{(L)}, \tag{7}$$

where $\leftarrow$ indicates that the left-hand side is updated with the value from the right-hand side. The updated representation $h_i^{(L)}$ is then input into a separate linear layer for predicting the residue type.

### 3.5 Pre-Training and Inference

Following the pre-training methodology in [58], we select 15% of residues at random for prediction. If the $i$-th residue is selected, we manipulate the $i$-th token by replacing it with: (1) the [MASK] token 80% of the time, (2) a random residue type 10% of the time, and (3) leaving the i-th token unchanged 10% of the time. The final hidden state $h^{(L)}$ is then used to predict the original residue type using cross-entropy loss. For S3F, we avoid information leakage from surfaces by removing the top 20 closest surface points for each selected residue. During pre-training, the weights of the ESM-2-650M model are frozen, and only the GVP layers for structure and surface graphs are trainable. This strategy helps preserve sequence representations and improves pre-training efficiency. Our models are pre-trained on a non-redundant subset of CATH v4.3.0 dataset (CC BY 4.0 license) [30], which contains 30,948 experimental structures with less than 40% sequence identity. S2F and S3F are trained with batch sizes of 128 and 8, respectively, for 100 epochs on four A100 GPUs. The pre-training time for S2F and S3F are 9 hours and 58 hours, respectively.

During inference, we adopt a strategy similar to that used in ESM-1v [5]. For each mutation, we mask the residue type at all mutation sites and remove the corresponding surface points, similar as the pre-training process. We use AlphaFold2 to predict the wild-type structures. To mitigate the influence of low-quality structures, for mutations on residues with a predicted Local Distance Difference Test (pLDDT) score below 70, we use the output scores from the baseline model, ESM-2-650M. For residues with pLDDT score no less than 70, we use the outputs from our S2F or S3F models.

## 4 Experiment

### 4.1 Setup

**Evaluation Dataset.** To assess the performance of our method on zero-shot protein fitness prediction, we run experiments on the ProteinGym benchmark (MIT License) [31]. This benchmark contains a comprehensive collection of Deep Mutational Scanning (DMS) assays, which covers a variety of functional properties such as thermostability, ligand binding, viral replication, and drug resistance. Specifically, we use 217 substitution assays that include both single and multiple mutations.

**Metric.** Given the complex, non-linear relationship between protein function and organism fitness [59], we select the Spearman's rank correlation coefficient as a primary metric for evaluating model prediction against experimental measurements. Besides, to ensure a comprehensive assessment, we include other metrics from the official ProteinGym benchmark: the Area Under the ROC Curve (AUC) and the Matthews Correlation Coefficient (MCC), which compare model scores with binarized experimental data. We also report the Normalized Discounted Cumulative Gains (NDCG) and Top 10% Recall with the aim to identify the most functionally effective proteins.

**Baseline.** We select a subset of baselines from the ProteinGym benchmark for comparison, categorizing them based on whether the model requires multiple sequence alignments (MSA) as input. In the category without MSAs, we include three protein language models—ProGen2 XL [50], CARP-640M [51], and ESM-2-650M [32]; three inverse folding models—ProteinMPNN [22], MIF [23], and ESM-IF [21]; and three sequence-structure hybrid models—MIF-ST [23], ProtSSN [27], and SaProt [26]. For models utilizing alignments, we choose three family-specific models—DeepSequence [6], EVE [7], and GEMME [8], as well as three hybrid models that combine family-agnostic and specific approaches, including MSA Transformer [13], Tranception L with retrieval [14], and TranceptEVE [15]. All baseline results are taken directly from the ProteinGym benchmark [31].

Table 1: **Overall Results on ProteinGym.** Models are categorized into two groups based on their reliance on MSA inputs. The types of input information and the number of trainable parameters are listed for each model. The best models within each category are highlighted in **red**.

| Model | Benchmark Results | | | | | Model Information | | | | |
|---|---|---|---|---|---|---|---|---|---|---|
| | Spearman | AUC | MCC | NDCG | Recall | Seq. | Struct. | Surf. | MSA | # Params. |
| ProGen2 XL | 0.391 | 0.717 | 0.306 | 0.767 | 0.199 | | | | | 6,400M |
| CARP | 0.368 | 0.701 | 0.285 | 0.748 | 0.208 | ✓ | ✗ | ✗ | ✗ | 640M |
| ESM2 | 0.414 | 0.729 | 0.327 | 0.747 | 0.217 | | | | | 650M |
| ProteinMPNN | 0.258 | 0.639 | 0.196 | 0.713 | 0.186 | | | | | 2M |
| MIF | 0.383 | 0.706 | 0.294 | 0.743 | 0.216 | ✗ | ✓ | ✗ | ✗ | 3M |
| ESM-IF | 0.422 | 0.730 | 0.331 | 0.748 | 0.223 | | | | | 142M |
| MIF-ST | 0.383 | 0.717 | 0.310 | 0.765 | 0.226 | | | | | 643M |
| ProtSSN | 0.442 | 0.743 | 0.351 | 0.764 | 0.226 | ✓ | ✓ | ✗ | ✗ | 148M |
| SaProt | 0.457 | 0.751 | 0.359 | 0.768 | 0.233 | | | | | 650M |
| **S2F** | 0.454 | 0.749 | 0.359 | 0.762 | 0.227 | ✓ | ✓ | ✗ | ✗ | 6M |
| **S3F** | **0.470** | **0.757** | **0.371** | **0.770** | **0.234** | | | ✓ | | 20M |
| DeepSequence | 0.419 | 0.729 | 0.328 | 0.776 | 0.226 | | | | | 70M |
| EVE | 0.439 | 0.741 | 0.342 | 0.783 | 0.230 | ✓ | ✗ | ✗ | ✓ | 240M |
| GEMME | 0.455 | 0.749 | 0.352 | 0.777 | 0.211 | | | | | <1M |
| MSA Transformer | 0.434 | 0.738 | 0.340 | 0.779 | 0.224 | | | | | 100M |
| Tranception L | 0.434 | 0.739 | 0.341 | 0.779 | 0.220 | ✓ | ✗ | ✗ | ✓ | 700M |
| TranceptEVE L | 0.456 | 0.751 | 0.356 | 0.786 | 0.230 | | | | | 940M |
| **S2F-MSA** | 0.487 | 0.767 | 0.381 | 0.790 | 0.240 | ✓ | ✓ | ✗ | ✓ | 246M |
| **S3F-MSA** | **0.496** | **0.771** | **0.387** | **0.792** | **0.244** | | | ✓ | | 260M |

*Since the numbers of trainable parameters for family-specific models depend on the length of protein sequences, we use the average length (400 AAs) of all ProteinGym sequences for estimation.

We report the zero-shot performance of our methods, S2F and S3F. To benchmark against alignment-based models, we further enhance our models by ensembling them with EVE predictions through the summation of their z-scores, resulting in two variants named S2F-MSA and S3F-MSA.

## 4.2 Benchmark Result

We report the benchmark results and model information for all baselines in Table 1. Among the methods that do not require MSAs, S2F achieves competitive results, surpassing protein language models and inverse folding models, and only slightly lagging behind SaProt. When augmented with surface features, S3F becomes the best model in this category, even outperforming the top alignment-based model, TranceptEVE, by a significant margin in terms of Spearman's rank correlation. Despite relying on protein language models, our methods have substantially fewer trainable parameters compared to other baselines (20M in S3F *v.s.* 650M in SaProt). Hence, our methods finish pre-training within several days, whereas some large-scale baselines require months. Additionally, when enhanced with alignment information, S3F-MSA improve the current state-of-the-art method, SaProt, by 8.5% in terms of Spearman's rank correlation, further demonstrating the potential of our approach. These results highlight both the effectiveness and the parameter efficiency of our proposed methods.

## 4.3 Breakdown Analysis for Multi-Scale Learning

From Table 1, we observe the progressive improvements achieved by gradually incorporating various protein aspects into the model, as demonstrated by comparing the results of ESM2, S2F, S3F, and S3F-MSA. To better understand the impact of this multi-scale representation learning, we conduct a breakdown analysis on different types of assays. In Fig. 2(a-d), we report the performance of these four methods on assays grouped by function types, MSA depths, taxon, and mutation depths. Our analysis reveals the following key points:

**1. Overall:** Consistent improvements are observed across all types of assays in different categories when structure, surface, and alignment information are added to the model.

**2. Function type:** Introducing structure and surface features significantly enhances performance on binding and stability assays. This aligns with our intuition that binding and stability are closely related to structural features, suggesting that structure-based methods have an advantage in identifying

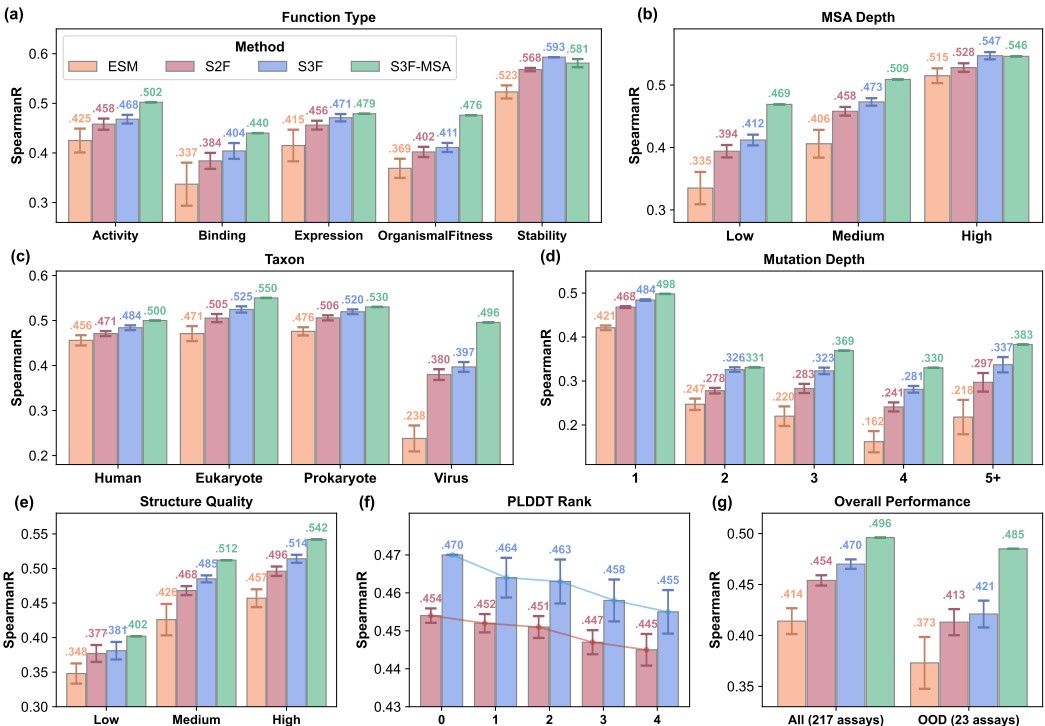

Figure 2: **Results of ESM-2-650M, S2F, S3F, and S3F-MSA for Analyzing Contributions of Sequences, Structures, Surfaces, and Alignments. (a-d)** Breakdown performance (Spearman's rank correlation) on assays grouped by function type **(a)**, MSA depth **(b)**, taxon **(c)**, and mutation depth **(d)**. **(e-f)** Impact of protein structure quality on performance. **(e)** Breakdown performance on assays with low, medium, and high-quality structures. **(f)** Results using five groups of AlphaFold2-predicted structures ranked by pLDDT (0 for the highest pLDDT, 4 for the lowest pLDDT). **(g)** Results on all assays and out-of-distribution assays with low sequence similarity to the pre-training dataset.

structure-disrupting mutations. This observation is consistent with previous studies [24], while our methods also maintain competitive performance across other function types.

**3. MSA Depth:** As shown in Fig. 2(b), protein language models perform poorly on assays with low MSA depths but excel on those with high MSA depths. This may be because proteins with low MSA depth are underrepresented in the ESM2 pre-training dataset, potentially reducing the diversity of specific families in protein language models. Introducing structure features partially mitigates this issue, and explicitly including family-specific training, like EVE, results in significant improvements.

**4. Taxon:** Leveraging structure and surface-based features consistently improves the fitness prediction performance across taxa. Critically, when the underlying protein language model is poor for a given taxon (eg., ESM on viral proteins [31]), incorporating structure and surface components provides inductive biases that help overcome these limitations. Developing taxa-specific models with higher quality node representations would likely results in further performance gains.

**5. Mutation Depth:** Most methods perform better on single mutations than on multiple mutations, likely due to our simplified additive assumption between mutations. As mutation depth increases, the performance gains from structure and surface encoding become more pronounced. This suggests that structure- and surface-based models are better at capturing epistatic effects.

In conclusion, multi-scale representation learning consistently improves performance. Incorporating structure and surface features can potentially correct biases in sequence-based methods, enhance accuracy in structure-related functions, and improve the ability to capture epistatic effects.

### 4.4 Impact of Structure Quality

To facilitate the usage of our methods on proteins without experimentally determined structures, we employ AlphaFold2 to generate structures for ProteinGym assays. By default, five structures are

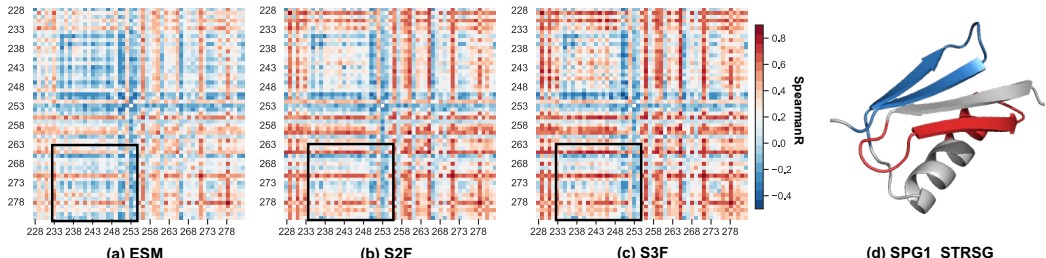

Figure 3: **Case Study on GB1. (a-c)** For each pair of mutation sites, we plot the Spearman's rank correlation between the experimental values and model-predicted scores for all 361 mutations: ESM (a), S2F (b), and S3F (c). The epistasis between residues 234-252 and residues 266-282 (in the black rectangle) are better captured by S2F and S3F. **(d)** Visualization of the predicted structure for GB1. Mutation regions 234-252 and 266-282 are highlighted in red and blue, respectively.

generated for each assay, with the best one selected based on pLDDT scores. Now we study how the quality of these predicted structures influences performance through two experiments.

First, we categorize the 217 assays into three groups based on the pLDDT of their predicted structures: 95 assays with pLDDT over 90 are classified as high-quality, 18 assays with pLDDT less than 70 as low-quality, and the remaining 104 assays as medium-quality. We report the average results for these categories in Fig. 2(e). The results indicate that performance for all four baselines positively correlates with structure quality, even for the sequence-based method, ESM. This may suggest our limited prior knowledge about these proteins with low-quality structures for all models, including AlphaFold2 and ESM2. Additionally, while improvements from structure and surface features are observed across all assays, those with high-quality structures benefit more significantly, especially for S3F. This underscores the importance of accurate structures for fitness prediction.

Next, we analyze the impact of structure quality using the five AlphaFold2-predicted structures. We perform fitness predictions five times, each time using a different group of structures ranked by their pLDDT scores. Specifically, we test S2F and S3F using the top 1, 2, 3, 4, and 5 predicted structures for all assays, respectively. The results, plotted in Fig. 2(f), show clear performance drops when lower-quality structures are used. This further highlights the reliance on high-quality structures.

## 4.5 Generalization Ability to Unseen Protein Families

In contrast to large-scale models pre-trained on UniRef100 or the AlphaFold Database, our method employs a much smaller dataset, CATH, which contains only 31,000 structures after clustering. This raises the question of whether the benefits of structure- and surface-based methods can generalize well to unseen protein families, despite the limited scale of the pre-training data. To address this, we select 23 out-of-distribution assays from ProteinGym, whose sequences have less than 30% similarity to the pre-training dataset. We plot the average performance of the four methods on these assays in Fig. 2(g). Although the absolute performance of all four methods decreases compared to their overall performance, we still observe consistent improvements from the structure- and surface-based methods, as evidenced by the performance gains of S2F over ESM and S3F over S2F. This proves the generalization ability of our methods to unseen protein families.

## 4.6 Case Study: Epistatic effects in the IgG-binding domain of protein G

We illustrate the advantages of our method by focusing on the results from the high-throughput assay of the IgG-binding domain of protein G (GB1) introduced in [60]. The assay quantifies the effects of all single and double mutations, thereby providing detailed insights into the epistatic relationships between residue pairs in the domain. We report in Fig. 3 the pair-specific Spearman's rank correlation between the experimental measurements and predictions from ESM, S2F and S3F, respectively. Leveraging structure and surface features leads to a superior ability in predicting mutation effects – in particular the epistatic effects between residue pairs that are far in sequence space but close in the tertiary structure of GB1 (off-diagonal terms in Fig. 3(a-c) and colored residues in Fig. 3(d)).

# 5 Conclusion and Limitations

This work introduces S3F, a novel multi-scale representation learning framework for protein fitness prediction that achieves state-of-the-art performance on the ProteinGym benchmark with a lightweight model. The breakdown analysis provides insights into how different protein modalities contribute to predicting various fitness landscapes. However, there are still several limitations in this work. First, our models are pre-trained on relatively small set of experimentally determined protein structures and may further benefit from leveraging more diverse structure sets, such as the AlphaFold database. Second, we make simplifying assumptions, such as ignoring side-chain information and assuming that backbone structures remain unchanged after mutations, which may limit the model's capacity. Third, our approach is limited to substitutions effects, and does not handle insertions nor deletions.

## Acknowledgments

The authors would like to thank Jiarui Lu and Chenqing Hua for their helpful discussions.

This project is supported by AIHN IBM-MILA partnership program, the Natural Sciences and Engineering Research Council (NSERC) Discovery Grant, the Canada CIFAR AI Chair Program, collaboration grants between Microsoft Research and Mila, Samsung Electronics Co., Ltd., Amazon Faculty Research Award, Tencent AI Lab Rhino-Bird Gift Fund, a NRC Collaborative R&D Project (AI4D-CORE-06) as well as the IVADO Fundamental Research Project grant PRF-2019-3583139727.

P.N. was supported by a Chan Zuckerberg Initiative Award (Neurodegeneration Challenge Network, CZI2018-191853). D.M. holds a Ben Barres Early Career Award from the Chan Zuckerberg Initiative as part of the Neurodegeneration Challenge Network (CZI2018-191853) and is supported by a NIH Transformational Research Award (TR01 1R01CA260415).

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

Table 2: **Average Spearman and Std. Error of Difference to Best Score.** The best performing model is S3F-MSA, highlighted in red. Δ Spearman is the difference between the baseline models and the S3F-MSA. We followed the ProteinGym benchmarks to compute the std. error of difference to best score. In this table, it is the bootstrapped std. error of the differences between the Spearman of each model and the Spearman of S3F-MSA.

| Model | Avg. Spearman | Δ Spearman | Std. Err of Diff. to Best Score |
|---|---|---|---|
| ProGen2 XL | 0.391 | -0.105 | 0.010 |
| CARP | 0.368 | -0.128 | 0.013 |
| ESM2 | 0.414 | -0.082 | 0.013 |
| ProteinMPNN | 0.258 | -0.238 | 0.013 |
| MIF | 0.383 | -0.113 | 0.011 |
| ESM-IF | 0.422 | -0.074 | 0.010 |
| MIF-ST | 0.383 | -0.113 | 0.010 |
| ProtSSN | 0.442 | -0.054 | 0.007 |
| SaProt | 0.457 | -0.039 | 0.000 |
| **S2F** | 0.454 | -0.042 | 0.005 |
| **S3F** | 0.470 | -0.026 | 0.005 |
| DeepSequence | 0.419 | -0.077 | 0.007 |
| EVE | 0.439 | -0.057 | 0.005 |
| GEMME | 0.455 | -0.041 | 0.006 |
| MSA Transformer | 0.434 | -0.062 | 0.011 |
| Tranception L | 0.434 | -0.062 | 0.007 |
| TranceptEVE L | 0.456 | -0.040 | 0.006 |
| **S2F-MSA** | 0.487 | -0.009 | 0.001 |
| **S3F-MSA** | **0.496** | 0.000 | 0.000 |

# A  Broader Impact

The main objective of this project is to model protein fitness landscape more accurately with multi-scale representation learning. Our approach utilizes structural information in the CATH dataset to build a multi-scale encoder with both sequence, structure and surface features. This advantage allows for more comprehensive analysis of protein research and holds potential benefits for various real-world applications, including protein engineering, sequence and structure design. It is important to acknowledge that powerful fitness prediction models can potentially be misused for harmful purposes, such as the design of dangerous drugs. We anticipate that future studies will address and mitigate these concerns.

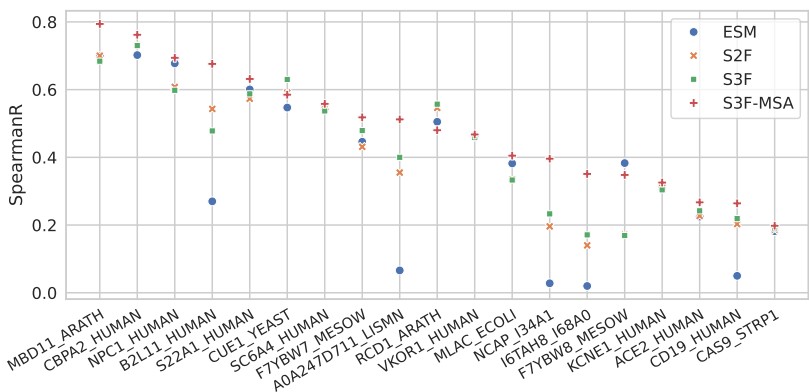

Figure 4: Spearmanr's rank correlation for ESM-2-650M, S2F, S3F and S3F-MSA on 19 proteins with less than 30% sequence similarity to the pre-training dataset.

# B   Additional Experimental Results

## B.1   Performance on OOD test sets

In Section 4.5, we select 23 out-of-distribution assays from ProteinGym by filtering with sequence similarity. Here we report the performance on these assays in Figure 4. The results on assays with the same proteins are grouped together.

## B.2   Statistical Significance Analysis

To quantify the statistical significance of the performance, we follow the same methodology as in ProteinGym and compute the non-parametric bootstrap standard error of the difference between the Spearman performance of a given model and that of the best overall model (10k bootstrap samples). Our performance delta with prior methods are all statistically significant, as shown in Table 2.

## B.3   Ablation Study

We provide ablation study results in Table 3, confirming the performance lift from the various modalities involved. The ablation dropping both structure and surface features corresponds to ESM2 in Table 1. Note that surface message-passing is designed to capture fine-grained structural aspects that enhance the coarse-grained features learned by our S2F (sequence+structure) model. However, relying solely on these fine-grained features without the context from structural features, as we do in the ablation removing structural inputs, appears to be detrimental to performance.

Table 3: Ablation Study for S3F.

| Model | Spearman |
|---|---|
| S3F w/o structure & surface | 0.414 |
| S3F w/o structure | 0.392 |
| S3F w/o surface | 0.454 |
| **S3F** | **0.470** |

