# OpenReview forum: "Multi-Scale Representation Learning for Protein Fitness Prediction"
_NeurIPS.cc/2024/Conference — NeurIPS 2024 poster_

### Official Review · Reviewer_6ssw · 2024-06-14

**Soundness:** 3
**Presentation:** 3
**Contribution:** 2
**Rating:** 5
**Confidence:** 5

**Summary:**

This work targets at the protein fitness prediction and introduces a sequence-structure-surface multi-modality (aka. multi-scale) self-supervised learning scheme. The results show that S3F outperforms baseline algorithms and achieves SOTA in the ProteinGym benchmark. I like the inspiration and favor S3F's promising performance. However, the study is very close to some recently published papers at ICLR 2024 (i.e., ProteinINR and PPIFormer) but completely ignores them. A fair comparison and analysis of the difference is necessary and I am afraid this is a big reason for me to give a borderline score.

**Strengths:**

(1) I advocate the research direction of pretraining on all feasible protein modalities, containing sequence, structure, and surfaces. Each representation form has its strengths and an incorporation is beneficial for the zero-shot protein fitness prediction.

(2) The author conducts extensive ablation studies over function type, MSA depths, taxon, and mutant depth. This helps readers to better understand the impact of each component of their model. The OOD examination of unseen protein families is also encouraging to demonstrate the superiority.

() In section 4.4, the author navigated the impact of structure quality and observed a clear performance drop when lower-quality structures were used. This phenomenon underscores the importance of accurate structures for fitness prediction. Notably, there have already been some studies that try to bridge the gap between representations in real and predicted structures [A], and I would recommend the author take a look if seeking further performance improvement.

[A] Protein 3D Graph Structure Learning for Robust Structure-Based Protein Property Prediction. AAAI 2024.

(3) The visualization and experimental analysis are elegant. I learned a lot from it.

**Weaknesses:**

(1) Missing of some closely related baselines and relevant work, GearNet [A] is one of the earliest studies in structural pretraining. ProteinINR [B] also leverages sequence, structure, and surfaces in a self-supervised way. The only difference is that they are not specifically designed to solve zero-shot problems. Based on these facts, the so-called "multi-scale representation learning" of S3F is no longer novel to me. From my point of view, both GearNet and ProteinINR can be simply adjusted to realize zero-shot prediction. Thus, it would be more interesting to see whether S3F outpasses them in all categories of downstream tasks.

Besides, there are also some appealing self-supervised methods to predict mutant effects. RDE [C] pretrain a structural encoder by masking and predicting side-chain angles. The author also did not discuss this line of research.

Last but not least, PPIFormer [D] pretrained a structural encoder with simple MLM in complex structures and can conduct zero-shot fitness prediction. I suppose it should also be compared and mentioned.

[A] Protein Representation Learning by Geometric Structure Pretraining. ICLR 2023.


[B] Pre-training Sequence, Structure, and Surface Features for Comprehensive Protein Representation Learning. ICLR 2024.

[C] Rotamer Density Estimator is an Unsupervised Learner of the Effect of Mutations on Protein-Protein Interaction. ICLR 2023.

[D] Learning to design protein-protein interactions with enhanced generalization. ICLR 2024.

(2) Some important relevant works are missing. For instance, the author claims that the weights of ESM-2 are frozen and only the structural encoder is tuned. This practice is very similar to the one in [A]. I would recommend the author at least cite it.

[A] Integration of Pre-trained Protein Language Models into Geometric Deep Learning Networks. Communications Biology 2023.


(3) The author used dMaSIF to generate the surface based on the backbone structures of proteins. This raises two doubts. Firstly, the side-chain atoms are ignored for surface generation, therefore the surface should be smaller than its standard one. Secondly, the fast-sampling algorithm developed by dMaSIF has unavoidable randomness. Different random seeds produce different surfaces. Has the author taken these two potential negative effects into consideration? Has the author adopted software like PyMol to acquire the surfaces?

**Questions:**

(1) In line 256, the author said "... by ensembling them with EVE predictions through the summation of their z-scores". I am not familiar with this alignment part. Can you please explain the details more?

(2) The title used "mutli-scale" reprsentations of proteins. However, from my personal point of view, multi-scale refers to atom-scale, residue-scale, and protein-scale, As this study proposed to leverage sequence, structure, and surfaces, "multi-modality" would be more appropriate.

(3) The author claimed that the backbone structures remain unchanged post-mutation. This is fine in most settings. However, did the author consider a more challenging circumstance, what if the structure varies significantly due to the mutation? [A] proposes a co-learning framework to simultaneously forecast the fitness change and the structural update. Does the author consider this factor?

[A] Thermodynamics-inspired Structure Hallucination for Protein-protein Interaction Modeling. ICLR submission.

(4) If I understand it correctly, the objective is to develop an unsupervised model that can predict a score for each mutant to quantify the changes in fitness values relative to the wild-type. Thus, it is more related to the mutant effect prediction task rather than the pure fitness prediction. I would recommend the author use "mutant effect prediction" instead of "fitness prediction" to better depict the target problem.

**Limitations:**

Yes

---

> ### Author Rebuttal · Authors · 2024-08-06
>
> Thank you for your insightful comments and suggestions. We have replied to each of your questions below and included additional analyses that we believe greatly strengthen our submission.
>
> >**C1: Missing of some closely related baselines and relevant work.**
>
> Thank you for pointing out additional recent baselines we could have considered. We follow your suggestion and compare against 4 additional baselines: ESM-GearNet, ESM-GearNet-Edge, RDE and PPIFormer. We could not include ProteinINR, since no public code is currently available for this work. We report the performance of these new baselines in the following table.
>
> Table A. Average spearman correlation over 217 ProteinGym assays
> |#Method|Spearmanr.|
> |:----:|:----:|
> |ESM-GearNet|0.412|
> |ESM-GearNet-Edge|0.432|
> |RDE|0.220|
> |PPIFormer|0.224|
> |S2F|0.454|
> |**S3F**|**0.470**|
>
> **Note that all the new baselines significantly underperform our suggested methods.** RDE and PPIFormer are specifically designed for predicting the mutational effects on protein-protein interactions, a task that fundamentally differs from protein fitness prediction. We had initially considered GearNet as our structure backbone in earlier stages of development, but eventually switched to GVP as it provided superior fitness prediction performance, as can be seen in the table above.
>
> >**C2: Some important relevant works are missing. For instance, the author claims that the weights of ESM-2 are frozen and only the structural encoder is tuned. This practice is very similar to the one in [A].**
>
> Thank you for the suggestion -- we will include this work in the background section of the final version.
>
> >**C3: There are two doubts about dmasif surface generation: (1) ignoring side-chain atoms makes the surface smaller. (2) randomness in surface generation. Has the author adopted software like PyMol?**
>
> Several great points! We answer your questions as follows:
> - Ignoring side-chain atoms indeed makes the surface smaller, but results in significant computational efficiency for surface generation and message passing. Our results demonstrate that we can achieve significant performance gains with limited computational overhead, making our implementation appealing to practitioners. If we set aside any computational consideration, an ideal modeling strategy would likely involve all-atom mutant structures and surfaces. We plan to explore such approaches in future work.
> - Randomness in surface generation is indeed unavoided. To mitigate this effect, we use a high-resolution surface (resolution = 1.0Å, with 6K-20K points) and construct a dense surface-to-backbone correspondence graph (using 20 nearest surface points). This approach ensures that our learned surface representations remain robust across different surfaces. Given your feedback, we tested S3F using surfaces generated with five different seeds and find that the standard deviation of the overall performance is approximately 0.001, thereby confirming the robustness of our learned surface features.
> - There are several good software options for surface generation, including PyMol. We chose dMaSIF as it offered efficient GPU implementation and could easily be integrated into our codebase, but other software suites could have been similarly used.
>
> >**C4: Can you please explain the details of the alignment part?**
>
> Since the model output from EVE are unnormalized delta ELBOs, model predictions from EVE and S3F are on unrelated scales. To facilitate model ensembling we thus first standard normalize model predictions separately, then take their arithmetic average. We will add this clarification in the revision.
>
> >**C5: "Multi-modality" would be more appropriate than “multi-scale” in the title.**
>
> Thank you for the suggestion. We will change the title accordingly in the final version.
>
> >**C6: Did the author consider a more challenging circumstance, what if the structure varies significantly due to the mutation? [A] proposes a co-learning framework to simultaneously forecast the fitness change and the structural update. Does the author consider this factor?**
>
> Thank you for bringing this work to our attention. The idea looks very interesting and related. We will reference this paper in the final version and consider the idea in the future work.
>
> >**C7: I would recommend the author use "mutant effect prediction" instead of "fitness prediction" to better depict the target problem.**
>
> We used the particular terminology used in the ProteinGym [1] work for consistency, although the two expressions are used synonymously in the relevant literature.
>
> [1] ProteinGym: Large-Scale Benchmarks for Protein Fitness Prediction and Design

---

> > ### Comment · Reviewer_6ssw · 2024-08-08
> > **Update**
> >
> > Thanks for your response. I am satisfied with the additional experiments and the outstanding performance compared to those important baselines. However, I still believe the way to generate surface using dMaSIF is not a smart choice despite their effectiveness in GPU parallel.
> >
> > However, I appreciate the efforts in answering my question and would like to raise my score to 5. I hope the authors can incorporate new changes during the rebuttal period into the final revision.

---

> > > ### Author Response · Authors · 2024-08-12
> > > **Concluding remarks**
> > >
> > > Dear reviewer,
> > >
> > > Thank you very much for reading through our responses and for raising your score. We will make sure to include the changes discussed above in the final revision.
> > > Please let us know if there are any other points we can clarify before the discussion period ends.
> > >
> > > Kind regards,
> > > The authors

---

### Official Review · Reviewer_rjUq · 2024-07-07

**Soundness:** 2
**Presentation:** 3
**Contribution:** 2
**Rating:** 5
**Confidence:** 3

**Summary:**

The paper presents a multimodal framework that integrates protein sequence, structure, and surface information together to predict protein fitness. The task of protein fitness prediction is a critical quality assessment of protein embeddings. Protein language models (pLM) are used for sequence representation which is embedded in a graph representation of protein structure and processed with geometric vector perceptron (GVP). The dMaSIF, a protein surface representation model, is used to encode protein surface, together with pLM embeddings of nearest residues. Surface features and sequence-structure features are integrated by concatenation and linear layers to predict masked residue identities. The model is evaluated on ProteinGym, a golden protein fitness benchmark dataset, against various pLM models and multiple sequence alignment (MSA) -based models and achieves favorable results in terms of spearman correlation with mutational effects. Evaluations are further run on AlphaFold predicted structures and less representative proteins to prove generalization.

**Strengths:**

**Originality**

This is one of the first papers that integrates pLMs, protein structure and surface representations together to predict protein fitness and shows improvement on benchmarks. It is also novel to combine MSA information with it to further improve predictive power of the model.

**Quality**

The submission is technically sound with a detailed description of their framework and extensive experimental results on benchmark datasets with appropriate reasoning and explanation.

**Clarity**

The paper is clear to read and easy to follow. Presentation of results are simple yet effective.

**Significance**

This method achieves SOTA performance on protein fitness prediction benchmark and provides a standardized way to integrate three modalities of protein together for a unified representation.

**Weaknesses:**

1. There is not adequate reference of prior work on similar topics, e.g. multimodal fusion and protein surface representation learning. The idea of integrating surface into protein representation is not novel, for example [1], but is neither acknowledged nor benchmarked in the paper.

2. Even though the performance of proposed method is better than competing methods, it's marginal. As the author writes in abstract '...these sequence-structure models have so far achieved only incremental improvements when compared to the leading sequence-only approaches.' However, SaProt (sequence+structure) provides 0.35 performance gain of correlation compared to ESM2 (seq) while S3F (proposed) provides 0.13 compared to SaProt. The advantage brought by surface representation is more incremental.

3. Statistical results should be provided for figure 2 when possible. How is the variation of performance on 217 assays?

References:
[1] https://arxiv.org/pdf/2309.16519

**Questions:**

1. line 195. ESM features are integrated into surface representations. Why not just use surface feature itself because it contains chemistry and geometry information already?

2. line 218. 'we avoid information leakage from surfaces by removing the top 20 closest surface points for each selected residue' How to ensure 20 is sufficient?

3. line 230. What percentage of results are output from ESM?

4. Figure 3. Why do some residues have negative correlation to model scores? Is there any quantitative analysis of differences rather than just qualitative visualization?

5. line 347. 'ignoring side-chain information'. If side chain is ignored, how to get surface information?

**Limitations:**

The majority of experiments in the paper is accomplished with experimental structure rather than predicted structures. And the results show that ordinary prediction quality could harm performance of the method. Thus, it is not clear if the model can be finetuned on AFDB to achieve better performance because of the quality of side-chain conformations in predicted structures.

---

> ### Author Rebuttal · Authors · 2024-08-06
>
> Thank you for your insightful comments and suggestions. We have responded to each of your questions below and included additional analyses that we believe significantly improve our submission.
>
> >**C1: The idea of integrating surface into protein representation is not novel, for example [1], but is neither acknowledged nor benchmarked in the paper.**
>
> We kindly refer the reviewer to our clarifications regarding the novelty and contributions from our work in our response to all reviewers above. Our background section (section 2) referenced several prior works that had already leveraged structure and surface features for general protein representation learning -- AtomSurf belongs to the same category, and we will include it as an additional reference in the revision. However, we would like to reiterate that we did not benchmark against any of these prior works leveraging surface features since none of them, including AtomSurf, supports (zero-shot) fitness prediction, which is the core task we focus on. The novelty of our work comes through the careful architecture design to best leverage structure and surface features to achieve state-of-the-art protein fitness prediction performance.
>
> >**C2: The advantage brought by surface representation is more incremental.**
>
> Please review our clarifications regarding the significance of our performance lift in our response to all reviewers above (point 2). The advantage brought by incorporating surface features in terms of fitness prediction performance is substantial: +0.02 Spearman between S2F and S3F across the 217 assays from ProteinGym. This is as much as the performance lift conferred by incorporating structural features (S2F vs ESM2). Together, structural and surface features provide a performance lift (+0.04 Spearman) that is comparable to factoring in epistasis vs not (Potts vs PSSM), which can hardly be characterized as incremental to fitness prediction performance.
>
> >**C3: Statistical results should be provided for figure 2 when possible. How is the variation of performance on 217 assays?**
>
> Thank you for suggesting this analysis. As discussed in our overall response to all reviewers, we compute the non-parametric bootstrap standard error for the difference in Spearman performance between a given model and the best overall model, using 10k bootstrap samples. We do this given the inherent variation in experimental noise and data quality across assays, which leads all models to consistently perform lower on certain assays and higher on others. By focusing on the standard error of the *difference* in model performance, we abstract away this assay-dependent variability, and provide a quantity that better reflects the statistical significance of the performance lift between our best model and any other baseline in ProteinGym. Please see the detailed results in the attached pdf, which confirms that the performance lift of our best model (S3F-MSA) vs the prior best baseline (SaProt) is statistically significant.
>
>
>
>
> >**C4: ESM features are integrated into surface representations. Why not just use surface feature itself because it contains chemistry and geometry information already?**
>
> We chose to combine ESM features with *geometric* features (such as Gaussian curvatures and Heat Kernel Signatures) to initialize the surface features. We believe that ESM can capture more informative chemical features by identifying residue types and co-evolutionary information, making it a strong complement to geometric  surface features.
>
> >**C5: How to ensure removing 20 closest surface points for each selected residue is sufficient?**
>
> We empirically chose this number of closest surface points to remove based on validation accuracy during pre-training. For instance, without removing these points, the pre-training accuracy quickly reaches 100%, suggesting likely information leakage. However, after removing these points, the S3F accuracy drops down to 52.4%, comparable to that of S2F (51.0%).
>
> >**C6: What percentage of results are output from ESM?**
>
> Out of 2.46 million mutations, approximately 0.35 million (14%) rely on ESM predictions only due to low-quality structures. We also benchmarked the results of S2F and S3F without pLDDT filtering, yielding aggregate Spearman performance scores of 0.449 and 0.461, respectively. These results remain significantly higher than the ESM performance (0.414 Spearman).
>
> >**C7: Why do some residues have negative correlation to model scores? Is there any quantitative analysis of differences rather than just qualitative visualization?**
>
> These negative correlations are already present in the ESM-based predictions, likely reflecting residue type preferences in ESM that do not align with experimental results. By introducing structural and surface features, our methods mitigate this effect to some extent. To quantitatively evaluate how much S2F and S3F improve over ESM methods, we calculated the average Spearman correlation in the regions of interest (residues 234-252 and 266-282). The results for ESM, S2F, and S3F are 0.301, 0.397, and 0.443 respectively, demonstrating that the introduction of structural and surface features can more effectively capture epistatic effects.
>
> >**C8: If the side chain is ignored, how to get surface information?**
>
> Since point mutations can significantly alter the side-chain structure, we chose to only keep the backbone structure for surface generation resulting in a smaller surface graph that is broadly applicable to all mutated sequences for the same protein family. This approach significantly reduces the cost of surface generation and message passing, making model training and inference more efficient, while yielding the significant performance improvement discussed above and keeping computations tractable on the hardware we had access to.

---

> > ### Comment · Reviewer_rjUq · 2024-08-11
> > **Reviewer response**
> >
> > Thank you for the response. If I understand correctly, the surface is calculated only from backbone structure? Then this definition is largely deviated from what people normally conceive as true protein surface which is mostly dictated by side chains. I believe the wording regarding surface is misleading and would cause confusion for audience interested in protein surface representation.

---

> > > ### Author Response · Authors · 2024-08-12
> > > **Concluding remarks**
> > >
> > > Dear reviewer,
> > >
> > > Thank you very much for reading through our responses and the additional question. Our surface features are indeed derived from the backbone structure only, which is both an approximation that has been used in prior literature (see for example [1]) and which has led to strong empirical performance in our experiments. We will further clarify this point in the revised manuscript, as well as include the rationale we had provided above for adopting this approach.
> > >
> > > Is there any other point of concern that we could help clarify before the discussion period ends?
> > >
> > > Kind regards,
> > > The authors
> > >
> > > [1] Hua et al. Effective Protein-Protein Interaction Exploration with PPIretrieval

---

### Official Review · Reviewer_6FTz · 2024-07-12

**Soundness:** 3
**Presentation:** 3
**Contribution:** 3
**Rating:** 6
**Confidence:** 4

**Summary:**

This paper proposes a new protein fitness prediction model, the Sequence-Structure-Surface (S3F) model, which integrates protein sequence information from a protein language model embedding, protein structure information processed through a Geometric Vector Perceptron (GVP) module, and protein surface information processed through the dMaSIF model and a GVP module. After pre-training S3F on CATH (to be specific, pre-training the GVP modules while freezing the protein language model weights), the model outperforms state of the art baselines on zero-shot protein function prediction. The paper also analyzes the breakdown of these results across different fitness prediction settings, finding that all settings benefit from adding structure, surface, or MSA information on top of sequence information, with the largest gains arising in binding and stability assays and in settings where protein language models have limited training data or biased training data.

**Strengths:**

The paper is well written and presents a parameter-efficient, novel way to include protein surface information into a model for fitness prediction. The paper contains results that will be interesting to the protein modeling community, showing that modeling the surface explicitly improves function prediction beyond other baseline methods which include structure information. It seems very interesting to investigate why such surface information is not easily captured by other methods that do utilize structure information.

**Weaknesses:**

1. The paper does not include results or analysis about the sensitivity of results for various hyperparameters of the model, such as the width, depth, and hidden dimension of GVP modules, the number of surface points to include in a neighborhood, and the choice of protein language model for embeddings. Since the paper focuses on the ProteinGym benchmark for its only evaluation, it would be nice to see more in-depth experimentation and/or ablation of different parts of the S3F model.
2. The central claim that surface information is important could also be strengthened with more experiments to understand how pre-processing structural information into surface features extracts useful information that is not as easily accessible with structure-based models that don’t do similar feature engineering/pre-processing.

**Questions:**

1. In Table 1, the bottom half of the table compares alignment-based models with S2F and S3F ensembled with EVE scores. It would be interesting to see results for each of the baselines in the top half of the table (e.g. MIF-ST, ProtSSN, SaProt) also ensembled with EVE scores.
2. What is your current intuition about why structure-aware fitness prediction models are unable to leverage surface information to its full extent?

**Limitations:**

The authors assess limitations of their work, including the fact that their method is limited to considering substitutions, and cannot handle insertions or deletions. I think this is a salient point to keep in mind, which limits the applicability of their method to real protein design tasks. See also weaknesses above.

---

> ### Author Rebuttal · Authors · 2024-08-06
>
> Thank you for your insightful comments and suggestions. We provide detailed responses to your questions below.
>
> >**C1: Lack of analysis for various hyperparameters of the model. It would be nice to see more in-depth experimentation and/or ablation of different parts of the S3F model..**
>
>
> During model development, we performed our main hyperparameter search based on the validation set accuracy for the S2F model (eg., number of layers, number of hidden dimensions), and used these same optimal values for S3F given the substantial compute costs required to process the surface features for pre-training. We provide below detailed ablation results for these model hyperparameters, as well as modality ablations. While the former tend to have a relatively marginal impact on the downstream performance, the latter clearly demonstrate the contributions of structural and surface information in S3F.
>
> Table A. Ablation Study and Hyperparameter Analysis.
> |#Method|Spearmanr.|
> |:----:|:----:|
> |S2F wider (512-dim)|0.446|
> |S2F deeper (8-layer)|0.457|
> |S3F w/o structure & surface|0.414|
> |S3F w/o structure|0.392|
> |S3F w/o surface (S2F)|0.454|
> |**S3F**|**0.470**|
>
> Note: we used 5 layers in our final S3F models, instead of the 8 layers that provide marginally better performance for S2F, due to GPU memory bottleneck when leveraging surface features.
>
>
>
>
> >**C2: The central claim that surface information is important could also be strengthened with more experiments to understand how pre-processing structural information into surface features extracts useful information that is not as easily accessible with structure-based models that don’t do similar feature engineering/pre-processing.**
>
> Surface message-passing is intended to capture fine-grained structural aspects that complement the coarse-grained features learned through structure message-passing. Additional ablations in Table A above clarify the respective benefits from including the different modalities.
>
> >**C3: In Table 1, the bottom half of the table compares alignment-based models with S2F and S3F ensembled with EVE scores. It would be interesting to see results for each of the baselines in the top half of the table (e.g. MIF-ST, ProtSSN, SaProt) also ensembled with EVE scores.**
>
> We report the ensembling results with these 3 baselines in the table below. We observe that S3F outperforms existing baselines in both regimes:
> When no MSA is available, S3F outperforms the original MIF-ST, ProtSSN and SaProt.
> When MSAs are available and we are willing to train protein-specific alignment-based models (eg. EVE) to increase fitness prediction performance, S3F-MSA leads to statistically significant higher performance over MIF-ST-MSA, ProtSSN-MSA and SaProt-MSA.
>
> Table B. Model Performance with EVE Ensembling.
> | #Method        | Spearmanr | Std. Error of Diff. to Best Score |
> |:----:|:----:|:----:|
> | MIF-ST-MSA    | 0.475         | 0.005                          |
> | ProtSSN-MSA   | 0.480         | 0.002                          |
> | SaProt-MSA    | 0.489         | 0.004                          |
> | **S3F-MSA**       | **0.496**         | **0.000**                          |
>
>
>
> >**C4: What is your current intuition about why structure-aware fitness prediction models are unable to leverage surface information to its full extent?**
>
> Explicitly including relevant information is a way to inject inductive bias into a task. For example, while structural information can be implicitly encoded in protein language models, we still want to use structure to augment PLM. Similarly, although surface information can be derived from protein structures, explicitly including it helps us better learn useful features for binding assays.

---

> > ### Comment · Reviewer_6FTz · 2024-08-11
> > **Thank you**
> >
> > Thank you to the authors for their additional experiments and careful response. I think the paper presents nice empirical results improving upon state-of-the-art ProteinGym performance, although the improvements are somewhat small (e.g. SaProt-MSA compared to S3F-MSA), and limited to that benchmark. Based on the responses and other reviewer comments, I will keep my score.

---

> > > ### Author Response · Authors · 2024-08-12
> > > **Concluding remarks**
> > >
> > > Dear reviewer,
> > >
> > > Thank you very much for reviewing our responses. As we understand it is the remaining point of concern, we would like to share a few concluding thoughts on the significance of our performance improvements. We consider the ProteinGym benchmark, with its 2.4 million mutation effect labels, to be the most critical benchmark for protein fitness prediction, akin to what ImageNet is for computer vision. While our experiments were focused on this benchmark, we believe it is the most relevant for the task we addressed. Regarding the improvement magnitude, we appreciate your feedback and would like to note the following:
> > > 1) SaProt-MSA was not an existing baseline but rather one we specifically computed for this rebuttal based on your suggestion. This result may offer valuable insights for the community, and we plan to include it in our revision
> > > 2) Even a ~1-point increase in Spearman correlation across 217 assays from ProteinGym represents a meaningful advancement, which we believe underscores the importance of leveraging additional data modalities, such as surface information, as we have proposed in this work.
> > >
> > > Please let us know if there are any other points we can clarify before the discussion period ends.
> > >
> > > Kind regards,
> > > The authors

---

### Official Review · Reviewer_ChRY · 2024-07-12

**Soundness:** 3
**Presentation:** 4
**Contribution:** 2
**Rating:** 5
**Confidence:** 4

**Summary:**

This paper augments a protein language model with two additional features, structure and surface information. The authors show that they can effectively use this information to predict protein function slightly better (though SOTA-level on relevant benchmarks) than sequence-only PLMs.

**Strengths:**

Significance: Perhaps the strongest point of the paper is that it clearly shows they can leverage structure and surface data to improve the zero-shot performance over sequence-only methods. The improvement is relatively small compared to models that only use sequence however.

Clarity: Paper is well written, has a good flow and logical steps.

Soundness: For the most part the right experiments are being done, the data benchmarks are correct and relevant models are tested (see weaknesses for a notable exception).

**Weaknesses:**

Novelty/Originality:
I can’t tell how this paper is a real advance on (https://arxiv.org/pdf/2204.02337, ref 33 in this paper), with the title being nearly identical too, which is published 2 years ago.

Yes the test set is different (evaluated on proteinGym), and some features of the architecture are also different, but the author choice of not benchmarking against Holoprot on the same dataset is problematic, if for nothing else because it’s the most clear way of showing the performance advance due to architectural improvement and no simply because of additional features (Because the “idea” of combining sequence, surface and structure is already executed.)

From the abstract
“Moreover, the function of certain 11 proteins is highly dependent on the granular aspects of their surface topology, 12 which have been overlooked by prior models. “
- Reference 33 (https://proceedings.neurips.cc/paper/2021/file/d494020ff8ec181ef98ed97ac3f25453-Paper.pdf) in your own citations is exactly about this, connecting sequence, structure and surface. Can you explain?
- Relatedly I think the similarity (almost identical : "Multi-Scale Representation Learning on Proteins") in the paper title is a poor decision, if not for the appearance of plagiarism, for the fact that it makes the difference between the papers even less clear. It should be titled to embolden the difference. That paper is also doing fitness prediction.

**Questions:**

I think the authors really need to contrast this work with reference 33 and argue why it’s a meaningful conceptual or performance advance.

I think the paper would be better if the authors would do at least two ablation studies where at least
- structure is dropped
- both structure and surface are dropped  (dropping sequence is more complex but it may also be interesting).

It is surprising to me that in the non MSA case S2F basically does not improve on benchmarks, and it could be that the structure data is adding no performance on top of the surface. Comparing to the original sequence-only model in this architecture would also help clarify how much improvement is simply due to other factors than surface information.

**Limitations:**

Conceptually the paper is doing something that has been done before, with some small innovations. It is relevant however that it can indeed consistently improve on existing language models.


:::::

Updating my previous score to 5

---

> ### Author Rebuttal · Authors · 2024-08-06
>
> Thank you for the very thoughtful comments and suggestions. We address each of your questions below, including several additional analyses which we believe significantly strengthen our submission.
>
> >**C1: The improvement is relatively small compared to models that only use sequence.**
>
> The best fitness prediction models using only sequence information are TranceptEVE [1] and GEMME [2] -- two methods leveraging multiple sequence alignments. Their aggregate Spearman performance on the 217 assays from the ProteinGym zero-shot substitution benchmark is of 0.456 for both (Table 1).
> In contrast, the performance of our best model (S3F-MSA) is 0.496. This is a massive performance lift (+0.04 Spearman), especially since averaged across 217 diverse DMS assays. To put things in perspective:
> This performance lift is larger than the performance lift between a PSSM and a Potts model (EVmutation) (0.359 vs 0.395) -- difference which can hardly be characterized as relatively small, given the critical importance of epistasis in protein fitness
> Over the past 5+ years, all 50+ fitness prediction baselines that were introduced after DeepSequence, the first deep learning-based approach for fitness prediction in 2018 [3], allowed to move from 0.419 to 0.457 (SaProt) aggregate Spearman. This also represents +0.04 Spearman performance lift -- our work extends over that best baseline by an extra +0.04 Spearman.
>
> Finally, we note that the current best baseline on ProteinGym that combines sequence and structure features is SaProt, with an 0.457 aggregate Spearman. The corresponding performance lift over the best sequence-only methods (+0.001 Spearman) *is* marginal, illustrating the complexity of effectively integrating new data modalities to boost zero-shot fitness prediction performance.
>
> >**C2: Novelty of the paper compared with HoloProt.**
>
> We kindly refer the reviewer to our clarifications regarding the novelty and contributions from our work in our response to all reviewers above. In particular, we would like to reiterate that, while our work is not the first to leverage structure and surface features for protein *representation learning*, it is the first to focus on and support *zero-shot protein fitness prediction*.
>
> This is a significant difference as methods developed for general protein representation learning do not enable zero-shot fitness prediction or, if they do, typically underperform methods specifically developed for it [4].
> For instance, in the work introducing HoloProt, Somnath et al. describe an approach for general protein representation learning using structure and surface features. The authors show that these representations can then be leveraged (via supervision) for several downstream tasks -- namely,  ligand binding affinity prediction and an Enzyme-Catalyzed Reaction Classification (which the authors refer to as “function prediction” in their abstract). Nothing in the work from Somnath et al. covers fitness prediction, let alone in the zero-shot context, and it is not clear how one could use the corresponding architecture to address the task that we focus on in our work.
>
> >**C3: Ablation study for the effect of structure and surface.**
>
> Thank you for the suggested ablation analyses. We provide the corresponding results in the table below, confirming the performance lift from the various modalities involved.
>
> Table A. Ablation Study.
> |#Method|Spearmanr.|
> |:----:|:----:|
> |S3F w/o structure & surface (ESM2)|0.414|
> |S3F w/o structure|0.392|
> |S3F w/o surface (S2F)|0.454|
> |**S3F**|**0.470**|
>
> Notes: 1) The ablation dropping both structure and surface features was already in our original submission, and corresponds to just using the underlying pLM (ie., ESM2). 2) Surface message-passing is designed to capture fine-grained structural aspects that enhance the coarse-grained features learned by our S2F (sequence+structure) model. However, relying solely on these fine-grained features without the context from structural features, as we do in the ablation removing structural inputs, appears to be detrimental to performance.
>
> >**C4: It is surprising to me that in the non MSA case S2F basically does not improve on benchmarks, and it could be that the structure data is adding no performance on top of the surface.**
>
> We believe this statement is inaccurate as the performance lift of S2F over ESM2 (the underlying pLM) is of +0.04 aggregate Spearman (0.454 vs 0.414), which, as we argue in our response to your first comment above (C1), corresponds to a transformational performance lift, on par with factoring in epistasis vs not, or commensurate with the performance lift obtained from over 5 years of deep learning literature for protein fitness prediction.
>
> [1] Notin, et al. "TranceptEVE: Combining family-specific and family-agnostic models of protein sequences for improved fitness prediction."
>
> [2] Laine, et al. "GEMME: a simple and fast global epistatic model predicting mutational effects." Molecular biology and evolution.
>
> [3] Riesselman, Adam J., John B. Ingraham, and Debora S. Marks. "Deep generative models of genetic variation capture the effects of mutations." Nature methods
>
> [4] Notin, et al. "Proteingym: Large-scale benchmarks for protein fitness prediction and design." NeurIPS.

---

> > ### Comment · Reviewer_ChRY · 2024-08-11
> > **Thank you**
> >
> > I thank the authors for their responses.
> >
> > Having read their answers, together with the other reviewers, I'm comfortable increasing my score. I do still feel like the conceptual advance on the previous paper and performance advance over the best benchmarks (given the auxiliary data it needs) is not at the level of a clear accept.

---

> > > ### Author Response · Authors · 2024-08-12
> > > **Concluding remarks**
> > >
> > > Dear reviewer,
> > >
> > > Thank you very much for your final feedback and for raising your score!
> > >
> > > As we near the end of the discussion period, we wanted to share a few concluding remarks regarding the novelty aspect of our work, as it appears to be the remaining point of concern. Firstly, we would like to emphasize that achieving the performance we obtained on ProteinGym required significant craftsmanship to optimally leverage the structural and surface features within our proposed architecture. In our view, this represents one of the many forms of novelty that NeurIPS aims to highlight. Secondly, our work is the first to explicitly demonstrate the value of these modalities for fitness prediction performance, offering another novel insight that we believe will be highly valuable for practitioners. Lastly, our method introduces a model-agnostic approach to augment protein language models with structural and surface features. This innovative aspect ensures that our approach can be seamlessly applied to enhance future protein language models as they continue to evolve and improve.
> > >
> > > Please let us know if there are any other points we can clarify before the discussion period ends.
> > >
> > > Kind regards,
> > > The authors

---

### Author Rebuttal · Authors · 2024-08-06

Dear reviewers,

We sincerely thank you for the time spent engaging with our paper and really appreciate the thoughtful comments. Based on your feedback, we have conducted additional experiments to further explore the strengths of our proposed approach, and have also clarified all points you had raised. We believe the submission is much stronger as a result. We summarize the key points of feedback and how we addressed them as follows:

1. **Novelty and contributions of this work (reviewers ChRY, rjUq, 6ssw)**
- The main criticism across reviews is the perceived lack of novelty over the prior protein representation learning works that had already introduced approaches to leverage protein structure and surface information. We would like to emphasize that all these prior works had been strictly focused on protein representation learning, and that none of them supports zero-shot protein fitness prediction -- which is the core task that our work focuses on. Thus, while methods to efficiently process protein structure & surface information are not novel, the most adequate way to use these modalities to obtain state-of-the-art protein fitness prediction performance is novel.
- Doing so is both non-trivial and of critical practical importance for the field of computational biology. It is important because fitness prediction is about understanding which proteins are functional and, as such, is one the most critical challenges underlying successful protein design: it is easy to generate novel proteins -- it is much harder to design functional ones. Additionally, zero-shot fitness prediction enables the quantification of mutation effects in settings where available experimental labels are scarce and/or difficult to collect (eg., environmental sensitivity, PTMs, allosteric regulation)
- Our work presents an effective approach to augment protein language models (eg., ESM) with structure and surface features: the underlying pLMs have mediocre zero-shot fitness prediction performance on their own, but our suggested approach endows them with state-of-the-art performance. Lastly, since our method is pLM-agnostic, we expect continuous progress in protein language modeling to yield even higher fitness prediction performance when combined with our approach.
2. **Significance of performance improvement (reviewers ChRY, rjUq, 6ssw)**
- We thoroughly evaluated our models against the 217 deep mutational scanning assays from the ProteinGym benchmarks. Our S3F and S3F-MSA models significantly outperform all 70+ baselines already present in ProteinGym, including recent top-performing models such as SaProt and ProtSSN (Table 1). As suggested by reviewers, we also compared against additional baselines, and significantly outperformed these as well (see next section).
- The overall performance increase (+0.04 Spearman) compared to the best baseline (SaProt) is substantial and comparable to significant modeling improvements, such as accounting for epistatic effects versus not (ie., the delta between a PSSM and a Potts model). For more details, please refer to our response to C1 from reviewer ChRY.
- To quantify the statistical significance of the performance, we follow the same methodology as in ProteinGym and compute the non-parametric bootstrap standard error of the difference between the Spearman performance of a given model and that of the best overall model (10k bootstrap samples). Our performance delta with prior methods are all statistically significant (see the attached pdf).
3. **New experiments conducted after reviews (all reviewers)**

Based on the feedback from all reviewers we conducted several new analyses as follows:
- Additional ablation keeping surface features but removing structure (reviewer ChRY; see C3): confirms the necessity to leverage both structure and surface features
- Additional hyperparameter results for our GVP (reviewer 6FTz; see C1): confirms optimality of chosen hyperparameters
- Sensitivity analysis wrt surface features (reviewer 6ssw; see C3): confirms performance is stable under various random seeds
- Additional baselines (reviewer 6ssw; see C1): confirms the superiority of our proposed architecture and the non-triviality in properly leveraging structure-based features for protein fitness predictions
- Statistical significance for fitness performance (reviewer rjUq; see C3): confirms performance deltas are all statistically significant

In addition to this overall response, we provide detailed responses to all comments raised by each reviewer. Please do reach out to us if you would like us to clarify any remaining points.

Thank you,

The authors

---

### Decision · Program_Chairs · 2024-09-25

**Decision:**

Accept (poster)

**Comment:**

The paper proposes a multimodal representation learning framework for protein by integrating structural and surface information, and applies it to zero-shot protein fitness prediction tasks. The paper addresses an important and practical task in the field of computational biology. Although the technical novelty of the method is relatively modest, this study could potentially benefit and inspire further research in protein studies. The reviewers have raised valuable comments, and the authors provide additional information and experiments in the rebuttal to address these concerns. Such content should be included in the revision to improve the paper quality and help better understand the contributions.